# Predicting the Distribution of the Invasive Species *Leptocybe invasa*: Combining MaxEnt and Geodetector Models

**DOI:** 10.3390/insects12020092

**Published:** 2021-01-21

**Authors:** Hua Zhang, Jinyue Song, Haoxiang Zhao, Ming Li, Wuhong Han

**Affiliations:** College of Geography and Environment Science, Northwest Normal University, Lanzhou 730070, China; zhao834323482@163.com (H.Z.); l1763650355m@163.com (M.L.); 2019212364@nwnu.edu.cn (W.H.)

**Keywords:** *L. invasa*, suitable growth area, MaxEnt, climate change

## Abstract

**Simple Summary:**

*Leptocybe invasa* is a global eucalyptus plantation invasive pest and the second alien invasive species in China. In this study, based on the current distribution data of *L. invasa* in China, combined with a geographic detector model and MaxEnt model, the main environmental variables were selected, and potential suitable growth areas of *L. invasa* in China in 2030 and 2050 were predicted. The results show that under the future climate change scenario, the potential distribution core areas of *L. invasa* in China will be located in Yunnan, Guangxi, Guangdong, and Hainan, and tend to spread to high latitudes (Hubei, Anhui, Zhejiang, Jiangsu, and other regions). Combined with the results of predicting the potential suitable zone in this study, we can clearly identify its diffusion trend, which has important theoretical significance for curbing the growth and development of *L. invasa* and formulating effective control measures.

**Abstract:**

*Leptocybe invasa* is a globally invasive pest of eucalyptus plantations, and is steadily spread throughout China. Predicting the growth area of *L. invasa* in China is beneficial to the establishment of early monitoring, forecasting, and prevention of this pest. Based on 194 valid data points and 21 environmental factors of *L. invasa* in China, this study simulated the potential distribution area of *L. invasa* in China under three current and future climate scenarios (SSPs1–2.5, SSPs2–3.5, and SSPs5–8.5) via the MaxEnt model. The study used the species distribution model (SDM) toolbox in ArcGIS software to analyze the potential distribution range and change of *L. invasa.* The importance of crucial climate factors was evaluated by total contribution rate, knife-cut method, and environmental variable response curve, and the area under the receiver operating characteristic (ROC) curve was used to test and evaluate the accuracy of the model. The results showed that the simulation effect of the MaxEnt model is excellent (area under the ROC curve (AUC) = 0.982). The prediction showed that *L. invasa* is mainly distributed in Guangxi, Guangdong, Hainan, and surrounding provinces, which is consistent with the current actual distribution range. The distribution area of the potential high fitness zone of *L. invasa* in the next three scenarios increases by between 37.37% and 95.20% compared with the current distribution. Climate change affects the distribution of *L. invasa*, with the annual average temperature, the lowest temperature of the coldest month, the average temperature of the driest season, the average temperature of the coldest month, and the precipitation in the wettest season the most important. In the future, the core areas of the potential distribution of *L. invasa* in China will be located in Yunnan, Guangxi, Guangdong, and Hainan. They tend to spread to high latitudes (Hubei, Anhui, Zhejiang, Jiangsu, and other regions).

## 1. Introduction

Alien invasive species refer to non-native species that migrate from their place of origin to a new ecological environment via natural or human-made means. They can reproduce in nature, resulting in damage to local biodiversity and impacts on the local ecological environment [1]. China suffers significantly from alien invasive species, and the annual economic loss caused by invasive species is as high as USD 119.876 billion [2]. To effectively prevent the harm caused by invasive species, it is essential to study their geographical distribution. The distribution of invasive species is affected by the interaction among species, the ability of species migration, climate, soil, hydrology, topography, and other factors, among which temperature is a critical factor [3]. The relevant report of the United Nations Intergovernmental Panel on Climate Change (IPCC) shows that, during the past 100 years, against the background of global warming, the global average surface temperature has increased by 0.85 °C, and the earth’s surface temperature will continue to show an upward trend in the future [4,5]. In the context of future global warming, the temperature in China will rise by 1.6 to 5.0 °C [6]. Many studies have shown that sustained climate warming may increase the tolerance of invasive species and expand their habitable areas, and they may form larger populations [7]. Therefore, studying the potential geographical distribution pattern of invasive species in the context of future climate change has important theoretical and practical significance for putting forward reasonable and effective biodiversity conservation measures.

Species distribution models (SDMs), also known as niche models, have been widely used to study the impact of climate change on the potential geographical distribution of species. The potential geographical distribution of species can be inferred from maximum temperature, minimum temperature, relative humidity, rainfall, and other environmental factors [8,9,10]. At present, commonly used niche models are the maximum entropy model (MaxEnt), the biological population growth model (CLIMEX), bioclimatic and domain models based on bioclimatic data, the niche factor analysis model (ENFA), and the genetic algorithm model (GARP) [11]. Among the many niche models, the MaxEnt model is the most widely used. The MaxEnt model was proposed by Phillips in 2004 [12]. According to the longitude, latitude, and environmental variable data of actual distribution points of a species, the prediction model obtained by a probability density operation is used to evaluate the possible distribution of target species in the target area. The MaxEnt model is widely used in the prediction of the suitable growth zone of a species because of its stable operation result, short operation time, and accurate prediction ability [13,14,15].

Eucalyptus is one of the fastest-growing tree species in the world, and plays an essential role in wood processing, papermaking, as a raw material, refining essential oils, etc., [16]. In China, the planting area of eucalyptus is more than 36.8 × 10^6^ km^2^, and the direct economic income is more than CNY 100 billion [17]. However, in line with the increase in the eucalyptus planting area, the number of eucalyptus pests has increased sharply [18]. The number of eucalyptus pest species in China increased from 53 in 1980 to 319 in 2011, resulting in a direct economic loss of more than CNY 1.125 billion per year [19]. *Leptocybe invasa Fisher & La Salle (Hymenoptera: Eulophidae)* was first discovered in Dongxing City [20], Guangxi PROVINCE in 2007 and was listed as the second alien invasive species in China by the Ministry of Ecology and Environment of the People’s Republic of China on 7 January 2010. This insect is a globally invasive pest in eucalyptus plantations, and can harm a variety of eucalyptus strains and form galls in various parts of the plant. In severe cases, it can cause seedling lodging and stop growth, causing substantial economic losses to the eucalyptus industry (e.g., *eucalyptus urophylla*, *corymbia polycarpa*, *southern mahogany*, and *queensland peppermint*) [21,22]. The early stages of adults and larvae of *L. invasa* are difficult to find, hampering early interception of the pests, which can therefore be readily introduced. In addition, small species can also promote their spread locally through unintentional transport. Strict quarantine measures may delay the spread of the *L. invasa*, but are unlikely to stop the invasion of all countries that grow eucalyptus [23]. The species originated in Australia but has subsequently appeared in 39 countries in Asia, Europe, Africa, and the Americas [24,25]. In China, in the space of a small number of years, *L. invasa* spread to the coastal areas of Guangdong, Fujian, and Hainan, and also migrated to Yunnan, Sichuan, Hunan, Jiangxi, and other regions in the inland areas. It has also spread to high latitude areas [26,27,28,29,30]. Therefore, it is essential to use the MaxEnt model to predict the potential distribution and migration route of *L. invasa* in China at present and in the future. Based on the current distribution data of *L. invasa* in China, combined with the geographic detector and MaxEnt models, this study screened the main environmental variables and predicted the potential suitable growth areas of *L. invasa* in China in 2030 and 2050.

The primary aims of this study are as follows: (1) To select the main environmental variables that affect the distribution of the species to establish a model; (2) to predict the spread of *L. invasa* according to SSPs1–2.5, 3.5, and SSPs5–8.5 climate change scenarios; (3) to analyze the potential suitable areas of *L. invasa* in China in 2000 and in the future (2030 and 2050), and to discuss the suitable growth range and future migration route of *L. invasa* affected by the main environmental factors affecting potential distribution. The research results provide an essential theoretical basis for the monitoring, early warning, and effective prevention and control of *L. invasa*.

## 2. Materials and Methods

### 2.1. Distributional Data

A comprehensive collection of *L. invasa*-related research articles, combined with the CABI database (http://www.cabi.org/cpc), global diversity information network (Global Biodiversity Information Facility), China National Health Pest Quarantine Information platform (http://www.pestchina.com/SitePages/Home.aspx), and China Forestry Pest [31] (results of the National Forestry Pest Survey 2014–2017) have recorded the location information of *L. invasa*. Using Google Earth (http://ditu.google.cn), these were translated into geographical coordinates. Finally, 194 accurate distribution records were obtained (Figure 1). The longitude and latitude coordinates of the sample were stored in an Excel database and converted into CSV format for the establishment of the MaxEnt model.

### 2.2. Environmental Variables and Processing

#### 2.2.1. Data Sources

Bioclimatic variables downloaded from the World Climate WorldClim2.0 Database (http://www.worldclim.org/) were used to build models for predicting species distribution. These variables include past climate data (1970–2000) and future climate data (2030: 2021–2040, 2050: 2041–2060) with a resolution of 2.5’. The National Climate Centre launched the earth system model BCC-ESM1.0 with an aerosol chemistry module, the medium resolution climate model BCC-CSM2-MR, and the high-resolution climate model BCC-CSM2-HR [32]. The BCCCSM2-MR climate model is widely used in Asia, particularly in China, so this model was used to predict the geographical distribution of *L. invasa*. The primary research on the future climate included four change scenarios (Table 1). Compared with the future climate change scenarios used in past studies, the shared socioeconomic pathways (SSPs) were launched by the Government Panel on Climate Change in 2010 [33]. SSPs have been used to quantitatively describe the relationship between climate change and socio-economic development paths. They play an increasingly critical role in predicting climate change and related research, and supporting climate policy decision-making.

#### 2.2.2. Geodetector Model

Geographic detectors are a set of statistical methods for detecting spatial differences and revealing the driving forces behind them. Their core idea is based on the assumption that if an independent variable has an essential influence on a dependent variable, then the spatial distribution of the independent variable and the dependent variable should be similar. Geographical detectors include factor detectors, interaction detectors, risk detectors, and ecological detectors. In this study, factor detectors were used to detect the extent to which environmental variables affect the spatial distribution of *L. invasa* [34]. The expressions are as follows:(1)$$q=1-\frac{\sum_{h-1}^{L}N_{h}s_{h}^{2}}{Ns^{2}}=1-\frac{SSW}{SST}$$
(2)$$SSW=\overset{L}{\underset{h=1}{\sum }}N_{h}\sigma_{h}^{2},SST=N\sigma^{2}$$
where *h* = 1, 2 …, *L*, is the stratification of variable Y or factor X; *N_h_* and *N* are the number of units in the layer *h* and the whole region, respectively; $\sigma_{h}^{2}$ and $\sigma^{2}$ are the variance of the Y value of the layer *h* and the entire area, respectively; *SSW* and *SST* are the sum of the total variance of the entire area. The *q* value represents the influence of environmental variables on *L. invasa* distribution, with a range pf [0, 1]. The higher the value, the stronger the effect on *L. invasa* distribution.

#### 2.2.3. Data Processing

Because too many environmental variables are not conducive to the prediction of the model, this study combined the knife-cutting method in the MaxEnt software (http://biodiversityinformatics.amnh.org/open source/Maxent, version 3.4.1) and the geographic detector software (http://www.geodetector.org/) to eliminate the environmental variables that make little contribution to the prediction results of the MaxEnt model. The specific operation steps of the geographic detector model were as follows: taking the species distribution point as the dependent variable and 21 environmental variables as independent variables, the 21 environmental variables were divided into five categories using the natural breakpoint method. Then, a 10 × 10 km fishing net with a total of 6729 points was used in ArcGIS10.5, and the dependent variable was matched with the independent variable via the fishing net. Factor detection analysis was carried out to obtain the size of the influence of each environmental variable (i.e., the q value, where a higher *q* value indicates greater influence of the factor) and the factor explanatory power value (the higher the *p* value, the smaller the explanatory power of the element). The environmental variables with a *q* value greater than 0.1 were screened out. Compared with the results of the knife cutting method in the MaxEnt software, 10 environment variables were selected for subsequent modelling (Table 2). The temperature and rainfall values of these bioclimatic variables were derived from each month and produced numerous biologically significant variables, which could be broadly divided into three categories: first, annual trends (e.g., annual mean temperature, annual precipitation); second, seasonality (e.g., annual temperature difference and precipitation); and third, extreme or restrictive environmental factors (for example, temperatures in the coldest and hottest months, and precipitation in wet and dry seasons). Because the comprehensive effects of a variety of bioclimatic variables can be considered, they are often used to predict species distribution and related ecological modeling techniques [35].

### 2.3. Species Distribution Model Evaluation

MaxEnt requires the user to specify a set of parameters, namely, the percentage of test training (i.e., the percentage of locations used for model development and internal testing), the number of background spots, the form of the functional relationship (the type of feature in the MaxEnt Language), clamp (i.e., whether to constrain the prediction within the variability of the input predictor), and the regularization multiplier (i.e., to avoid over-fitting of the response curve). However, there is no agreement in the literature on which set of parameter values to use in MaxEnt, and best practices recommend performing a preliminary sensitivity analysis of the performance of the parameters selected by the model. In this study, the lower area under the curve (AUC) value of the receiver operating characteristic (ROC) curve was used to test the accuracy of the results of the suitability analysis of *L. invasa*. The ROC curve is a receptivity curve, in which the abscissa represents the false positive rate (1—specificity), and the ordinate represents the true positive rate (1—omission rate) [12]. This analysis method was originally used for the analysis and monitoring of radar signals (the range of the AUC value is 0:1). The grade of simulation prediction is poor (AUC ≤ 0.80, 0.80 < AUC < 0.90), better (0.90 < AUC < 0.95), and excellent (0.95 < AUC < 1.00) [36]. Because the AUC value is not affected by the threshold, its evaluation of the model is more objective. The AUC value shows that it can separate the existence of the local distribution, which shows that the prediction effect of the MaxEnt model is better. However, although we used AUC, which is probably the most popular method to evaluate the accuracy of the predictive distribution model [37,38], to evaluate the performance of parameter configuration, we chose the parameter configuration in view of the controversy among scientists about its reliability. In this study, first, based on the known distribution points of *L. invasa* and its corresponding environmental variables, the RM was set to 0.5–4. Six feature combinations (FC) were used to optimize the model parameters to select the optimal parameter combination: L (linear features); LQ (linear features + quadratic features), H (hinge features), LQH (linear features + quadratic features + Hinge features), LQHP (linear features + quadratic features + hinge features + product features); and LQHPT (linear features + quadratic features + hinge features + product features + threshold features). Finally, the RM of this study was set to 1, the feature combination was LQHPT, and the proportion of the data of the distribution points of the verification set was 25%. The software randomly selects 75% of the data from the known distribution points of *L. invasa* as the training set using the cross-validation method (that is, the species distribution data are randomly divided into 10 parts, one of which is selected as the test set, and nine are selected as the training set). The maximum number of iterations is 500 and the maximum number of background attractions is 10,000. The knife-cutting method (jackknife test) was selected to determine the weight of each variable affecting the suitable growth area of *L. invasa*, and the environmental variable ROC was selected.

The final result was in ASCII format, which was mapped to the map of China after rasterization by Arcgis10.5 GIS software for further analysis. The grade of potential suitable zone was as follows: unsuitable habitat < 0.1; 0.1 ≤ poorly suitable habitat < 0.3; 0.3 ≤ moderately suitable habitat < 0.6; and highly suitable habitat ≥ 0.6.

### 2.4. Spatial and Statistical Analysis

ArcGIS 10.5 software (Esri, CA, USA) was used to calculate the areas of different suitable zones in different periods, and SDM toolbox 2.4 (Esri, CA, USA) was used to calculate changes in the potential distribution areas and distribution center of *L. invasa* in China in different periods [39]. Using the function of “Reclass” in ArcGIS 10.5 software, the grid values of the suitable and non-suitable zones predicted for *L. invasa* in each period were modified to 1 and 0, respectively. Then, the SDM toolbox was added to select the “MaxEnt Tools” subdirectory in the “SDM Tools” module. The “Distribution Changes Between Binary SDMs” tool was used to calculate the area variation range of potential distribution areas in each period (2030: SSPs1–2.6, SSPs 2–4.5, SSPs 5–8.5, 2050: SSPs1–2.6, SSPs 2–4.5, and SSPs 5–8.5), and the expansion region, stable area, and contraction area of the distribution were obtained. The “Centroid Changes (Lines)” tool (Esri, CA, USA) was used to calculate the geometric center displacement of the predicted distribution in different periods, to detect the overall change trend of the *L. invasa* distribution area, and to obtain the vector overlap density of the geometric center change. The high-density area of vector overlap is likely to be a crucial area in the process of species migration to suitable habitats.

### 2.5. Analysis of Multivariate Environmental Similarity Surface

The multivariate environmental similarity surface (MESS) was used to analyze the degree of ecological change of *L. invasa* in the distribution area under the future climatic background. MESS analysis first determines the reference layer of bioclimatic variables. It then calculates the similarity between bioclimatic variables under different climatic conditions and the point set of bioclimatic variables in the reference layer (similarity, S). When the S value is positive, the smaller the S value, the more significant the climate difference at the point; when the S value is 100, no difference exists; when the S value is negative, the S value of at least one bioclimatic variable at the point is beyond the reference range. The environmental change at this point is excellent [40]. This operation was implemented by running the “density.tools.Novel” tool in the maxent.jar file in the command window.

## 3. Results

### 3.1. Evaluation of MaxEnt Model Prediction Accuracy

Based on 194 current distribution records and ten environmental variables, the potential geographical distribution of *L. invasa* in China was simulated in MaxEnt software, in which the training AUC value was 0.982. The test AUC value was 0.976 (Figure 2), which represents an excellent level, indicating that the prediction result of the MaxEnt model is accurate and available, and has high predictability.

### 3.2. Main Environmental Factors Affecting the Distribution

The environmental factors that have significant influence on *L. invasa* distribution range calculated by MaxEnt model were as follows: annual mean temperature (Bio1) contribution rate was 43.4%; NDVI contribution rate was 20.0%; wettest seasonal rainfall (Bio16) contribution rate was 11.1%; lowest temperature (Bio6) contribution rate of the coldest month was 10.5%; and seasonal mean temperature (Bio11) contribution rate of the coldest month was 6.3%. The contribution rates of wettest quarterly average temperature (Bio8), annual temperature range (Bio7), warmest seasonal rainfall (Bio18), driest quarterly average temperature (Bio9), and isothermal (Bio3) were 3.7%, 2.1%, 1.2%, 1.0%, and 0.7% respectively (Figure 3). The results of the knife-cut test showed that when only a single ecological factor variable is used, the four environmental factor variables with the most significant influence were annual mean temperature (Bio1), lowest temperature of the coldest month (Bio6), driest seasonal mean temperature (Bio9), and coldest monthly seasonal mean temperature (Bio11) (Figure 4). Combining the two results, the main influencing factors were vegetation cover (NDVI), weather (annual average temperature, lowest temperature of the coldest month, driest seasonal average temperature, coldest seasonal average temperature), and precipitation (wettest seasonal rainfall). According to the response curve of the main environmental factors for *L. invasa* (Figure 5), when the probability of *L. invasa* distribution is ≥0.6, and the suitability grade is high suitable distribution, the normalized vegetation index is 0.10–0.64, the annual average temperature is 19.60–25.82 °C the coldest monthly minimum temperature is −33.78–17.50 °C, the driest seasonal average temperature is −28.40–22.77 °C, the coldest monthly average temperature is 8.13–18.82 °C, and the wettest season precipitation is 394.13–1946.78 mm. When the probability of *L. invasa* distribution reaches the maximum, the corresponding normalized vegetation index, annual average temperature, coldest month minimum temperature, driest seasonal average temperature, coldest month season average mild, and wettest season precipitation are 0.39, 25.82 °C, 17.50 °C, −10.65 °C, 9.80 °C, and 613.15 mm, respectively.

### 3.3. Current Potential Distribution

The current distribution map of the suitable growth area of *L. invasa* in China is shown in Figure 6. The results show that the total suitable area is about 43.91 × 10^4^ km^2^, accounting for 4.57% of the total land area of China, and the high, medium, and low suitable areas account for 9.02%, 24.12%, and 66.86% of the total area, respectively (Table 3). *L. invasa* suitable growth areas are mainly distributed in Yunnan Province, eastern Sichuan Province, Guangxi Province, Guangdong Province, Fujian Province, Hainan Province, and the western coastal areas of Taiwan Province. Among these, the area of high fitness area is about 3.96 × 10^4^ km^2^, which is mainly distributed in the south and southeast of Guangdong Province and the coastal areas of Fujian Province. A small number of broken distributions are also located in the north of Yunnan Province, the east of Sichuan Province, the south of Jiangxi Province, Guangxi Province, Hainan Province, and the northwest coastal area of Taiwan Province.

### 3.4. Potentially Suitable Climatic Distributions in the Future

The area of *L. invasa* in the next three emission scenarios will increase by a range of 37.37% to 95.20% compared with the current potential distribution of high growth zones in China (Table 3). Under the system of SSPs1–2.6, the total area of *L. invasa* in 2030 is 63.62 × 10^4^ km^2^, which represents an increase of 44.89% compared with the current distribution area; the areas of high, medium, and low habitats increase by 37.37%, 39.28%, and 47.92%, respectively. In 2050, the total suitable area of *L. invasa* is 71.78 × 10^4^ km^2^, which is 63.47% higher than the current distribution area; the area of high suitable area decreases by 5.30%, the area of medium suitable area increases by 58.17%, and the area of low suitable area increases by 74.66%. Under the scenario of SSPs2–4.5, the total area of *L. invasa* in 2030 is 69.28 × 10^4^ km^2^, which represents an increase of 57.78% compared with the current distribution area; the areas of high, medium, and low habitats increase by 42.68%, 48.54%, and 63.15%, respectively. In 2050, the total adaptive zone area of *L. invasa* is 88.85 × 10^4^ km^2^, which is 1.02 times higher than the current distribution area; the area of the high adaptive zone, middle adaptive zone, and low adaptive zone increase by 67.17%, 1.04 times, and 1.02 times, respectively. Under the scenario of SSPs5–8.5, the total suitable area of *L. invasa* in 2030 is 70.41 × 10^4^ km^2^, which represents an increase of 26.50% compared with the current distribution area; the areas of high, medium, and low growth increase by 50.50%, 53.26%, and 64.24%, respectively. In 2050, the total suitable area of *L. invasa* is 97.31 × 10^4^ km^2^, which is 1.23 times higher than the current distribution area; the areas of high, medium, and low growth increase by 95.20%, 1.33 times, and 1.21 times respectively (Figure 7).

As can be seen from Figure 8 and Figure 9 and Table 4, under the scenario of SSPs1–2.6, the distribution center of *L. invasa* in 2030 is located in Zhaoqing City, Guangdong Province, which is about 0.28° higher than that of the current distribution center. The increase in the distribution area is 17.68 × 10^4^ km^2^, the loss area is minimal, and the stable area is 40.74 × 10^4^ km^2^. The newly suitable areas mainly appear in the eastern part of the Sichuan Province, the southern part of Hunan Province, and the southern part of Jiangsu Province. Sporadic distribution exists in Yunnan, Guangxi, Guangdong, Fujian, and other provinces. Furthermore, the latitude of the *L. invasa* distribution center in 2050 is 1.37° higher than that in 2030, the increased area is 28.58 × 10^4^ km^2^, the loss area is 2.15 × 10^4^ km^2^, and the stable area is 38.60 × 10^4^ km^2^. The newly suitable areas mainly appear in Hunan Province, Jiangxi Province, the east of Hubei Province, and the east of Sichuan Province, and a small amount of broken distribution is also found in Anhui, Henan, Jiangsu, Shandong, and Hebei provinces. Under the scenario of SSPs2–3.5, the distribution center of *L. invasa* in 2030 is located in Wuzhou City, Guangxi Province, which is about 0.78° higher than the latitude of the current distribution center. The increased area of the distribution area is 22.80 × 10^4^ km^2^, the loss area is minimal, and the stable area is 40.73 × 10^4^ km^2^. The newly suitable areas mainly appear in the eastern part of Sichuan Province, the southern part of Hunan Province, and the northern part of Jiangxi Province, and sporadic distribution exists in Yunnan, Guangxi, Guangdong, Fujian, and Jiangsu provinces. The latitude of *L. invasa* distribution center in 2050 is 0.60° higher than that in 2030, the increased area is 40.95 × 10^4^ km^2^, the loss area is 0.05 × 10^4^ km^2^, and the stable area is 40.70 × 10^4^ km^2^. New suitable areas mainly appear in Hunan Province, Hubei Province, Jiangsu Province, Jiangxi Province, the west of Chongqing, and the east of Sichuan Province. Guizhou, Fujian, Zhejiang, Anhui, and Henan provinces also have a small amount of broken distribution. Under the scenario of SSPs5–8.5, the distribution center of *L. invasa* in 2030 is located in Wuzhou City, Guangxi Province, which is about 0.76° higher than the latitude of the current distribution center. The increased area of the distribution area is 24.08 × 10^4^ km^2^, the loss area is minimal, and the stable area is 40.73 × 10^4^ km^2^. The newly suitable areas mainly appear in the eastern part of Sichuan Province, the central region of Hainan Province, and the southern part of Jiangsu Province, and sporadic distribution exists in Hunan, Hubei, Jiangxi, and Fujian provinces. The latitude of the *L. invasa* distribution center in 2050 is 0.65° higher than that in 2030, and the increased area is 48.07 × 10^4^ km^2^, the loss area is minimal, and the stable area is 40.73 × 10^4^ km^2^. The newly suitable areas mainly appear in Guizhou Province, Hunan Province, Jiangxi Province, Jiangsu Province, the east of Hubei Province, and the east of Sichuan Province, and a small amount of broken distribution is also located in Anhui, Henan, Zhejiang, Guangdong, and Fujian provinces.

### 3.5. Analysis of the Multivariate Environmental Similarity Surface (MESS) of L. invasa Potential Areas of Distribution under Future Climate Changes

Under the future climate scenario, the area distribution of the climate anomaly area (S < 0, red area) in the whole potential distribution area is small (Figure 10). Compared with the potential distribution area under the same climate scenario in the same period in the future (Figure 7), the suitable growth area of *L. invasa* is not predicted in the climate anomaly area. Under the climate scenarios of 2030 SSPs1–2.6, 2030 SSPs2–3.5, 2030 SSPs5–8.5, 2050 SSPs1–2.6, 2050 SSPs2–3.5, and 2050 SSPs5–8.5, the average similarity values of the 194 modern effective distribution record points for *L. invasa* are 2.58, 2.46, 2.75, 1.76, 1.92, and 1.28, respectively, indicating that the degree of anomaly is higher in the 2030 SSPs5–8.5 climate scenario. The abnormal degree of the other five climate scenarios is low.

## 4. Discussion

The scope of invasion of invasive species in the future is difficult to predict. At present, scientists and scholars are using and relying on environmental variables (temperature and precipitation) to predict the potential distribution of alien invasive species and suitable habitats [13,41]. In this study, the MaxEnt model was used to predict the potential distribution range of branch gall wasp in 2030 and 2050 according to environmental variables. The AUC value tested by the MaxEnt model was 0.982, which shows that the prediction result of the model has high accuracy and credibility [36], and that the prediction result is also consistent with the actual distribution of *L. invasa* in China. According to the contribution rate of environmental factors to *L. invasa* and the importance of environmental factors in the model jackknife test, this study found that the temperature factor mid-year mean temperature (Bio1) is the most important factor affecting *L. invasa*. This is consistent with a study by Huang Rui, which found that the growth, development, reproduction, and distribution of *L. invasa* are mainly affected by temperature [42]. Chen Yuansheng et al. [43] measured the relationship between the growth and development of *L. invasa* and climate using indoor constant temperature inoculation and rearing. The results showed that the optimum temperatures for development were 25.68, 25.65, 24.58, 26.42, and 23.84 °C, and the optimum temperature for growth was 12.21–35.48 °C. In this study, the optimum range of annual average temperature of *L. invasa* was 19.60–25.82 °C. This is consistent with the optimal temperature predicted in the current model, which proves that our results are credible. In this study, when the distribution probability of *L. invasa* reached the maximum, the corresponding values of annual average temperature, coldest month minimum temperature, driest season average mild, and coldest monthly average temperature were 25.82, 17.50, −10.65, and 9.80 °C. Zhu Fangli et al. showed that the developmental zero temperature (DZT) of *L. invasa* in the whole life cycle (egg-adult) was about 19 °C, and the DZTs in egg, larva, and pupa are 13, 19.7, and 17 °C, respectively [44]. These temperatures are acceptable in this study.

Some studies have shown that, due to the influence of global climate change, the continuous rise of global temperature, and future changes of precipitation pattern (time and space) and precipitation intensity [45], will lead to the trend of migration of many species to high latitudes and high elevations [46]. For example, Jia Dong et al. [47] predicted that the apple red constricted aphid would spread to high latitudes and Northeast China in the future. Jamal et al. [48] predicted that the small beehive beetle would spread to northern Africa and parts of Europe in the future. The future geometric distribution centers of *L. invasa* predicted in this study generally migrate to high latitudes, and the range of distribution gradually spreads to the northeast. New suitable zones will appear in Hubei, Anhui, Zhejiang, Jiangsu, and other regions. For invasive species, the distribution is affected not only by climatic conditions, but also by the species and distribution of natural enemies, host distribution, topography, and other factors. It may also be related to the unique characteristics of parthenogenesis, low-temperature resistance, generation overlap, and individual size of *L. invasa* [49]. In recent years, the phenomenon of global warming has intensified, which has changed the suitable environment of many species. It can be seen that climate change has a significant impact on the suitable habitat of species.

Three leading suggestions can be made to prevent *L. invasa* invasion and spread: (1) strengthen monitoring and timely early warnings. According to the occurrence law and bio-ecological characteristics of *L. invasa*, monitoring sites should be set up in eucalyptus planting areas. Timely investigation and monitoring should be carried out, especially on the seedling production base, the surrounding areas of the occurrence area, and the young eucalyptus forests and road protection forests on both sides of urban traffic trunk lines. (2) Varieties or clones with strong resistance should be given priority for afforestation. Eucalyptus varieties or strains resistant to *L. invasa* have been planted to improve the resistance to *L. invasa* and reduce its occurrence and harm. (3) Scientific prevention and control to prevent spread damage. For the areas where *L. invasa* occurs, positive and effective measures should be implemented to control its damage, reduce the economic losses caused by disasters, and effectively prevent or slow the speed of outward spread.

## 5. Conclusions

In this study, the MaxEnt model was used to model *Leptocybe invasa* 194 distribution points and ten environmental factors to predict the suitable growth area of *L. invasa* under current and future climatic conditions. The results showed that the AUC value tested by the MaxEnt model was more than 0.980, indicating that the prediction accuracy was very high. Under the background of global climate change, *L. invasa* survives in areas where the annual average temperature is 19.60–25.82 °C, the lowest temperature in the coldest month is −33.78–17.50 °C, the average seasonal temperature in the driest month is −28.40–22.77 °C, and the average temperature in the coldest month is 8.13–18.82 °C. The precipitation in the wettest season is 394.13–1946.78 mm. Under the three future climatic scenarios, the area of potential high growth zones of *L. invasa* in China increases and tends to spread to high latitudes, which will become more evident with the process of climate warming. *L. invasa* is the second alien invasive species in China. Combined with the results of predicting potentially suitable areas in this study, the diffusion trend can be identified, which has important theoretical significance for curbing the growth and development of *L. invasa* and formulating effective control measures.

## Figures and Tables

**Figure 1 insects-12-00092-f001:**
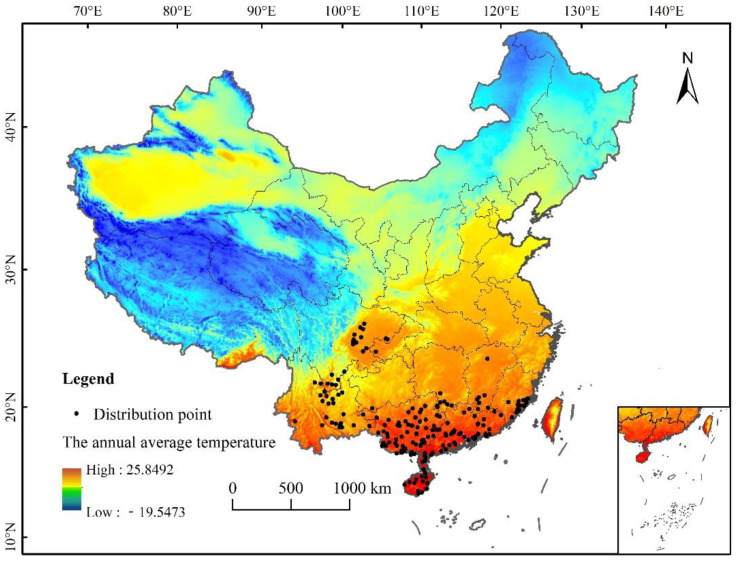
Distribution data of *L. invasa* in China.

**Figure 2 insects-12-00092-f002:**
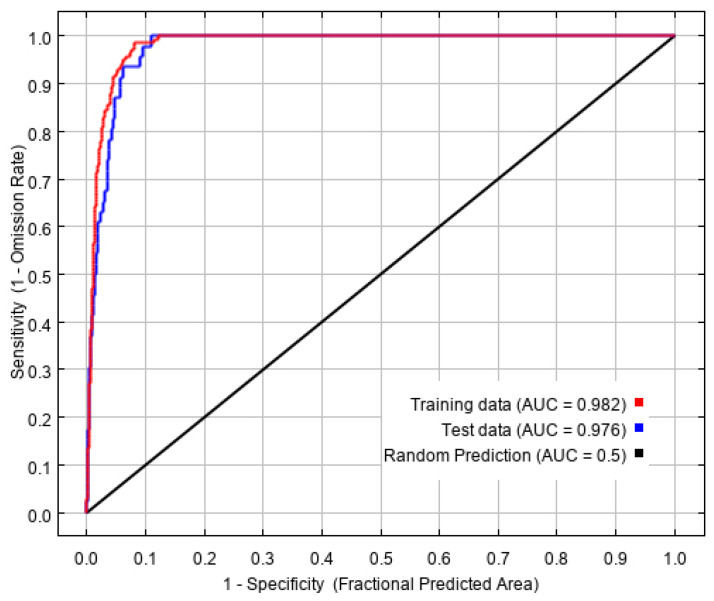
Reliability test of the distribution model created for *L. invasa*.

**Figure 3 insects-12-00092-f003:**
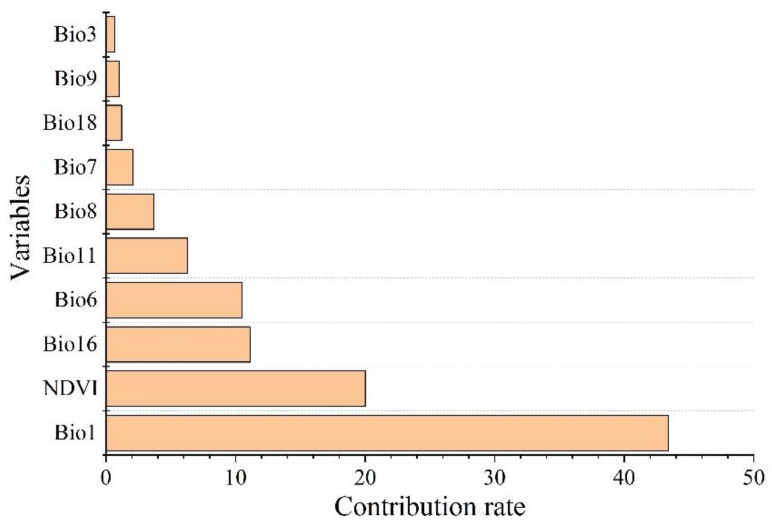
Permutation importance of variables for modeling.

**Figure 4 insects-12-00092-f004:**
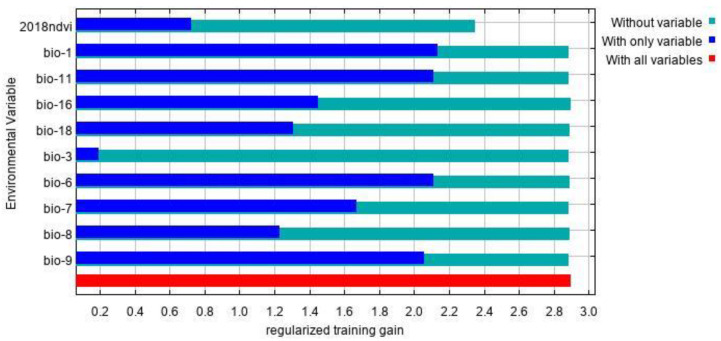
The jackknife test result of environmental factor for *L. invasa.*

**Figure 5 insects-12-00092-f005:**
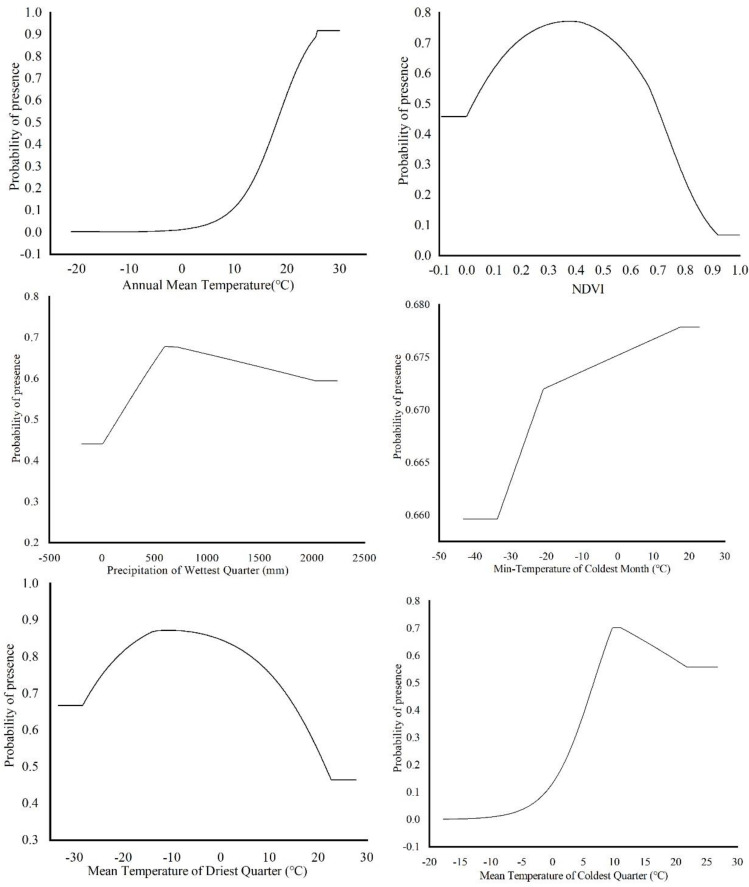
Response curves of probability of presence for *L. invasa*.

**Figure 6 insects-12-00092-f006:**
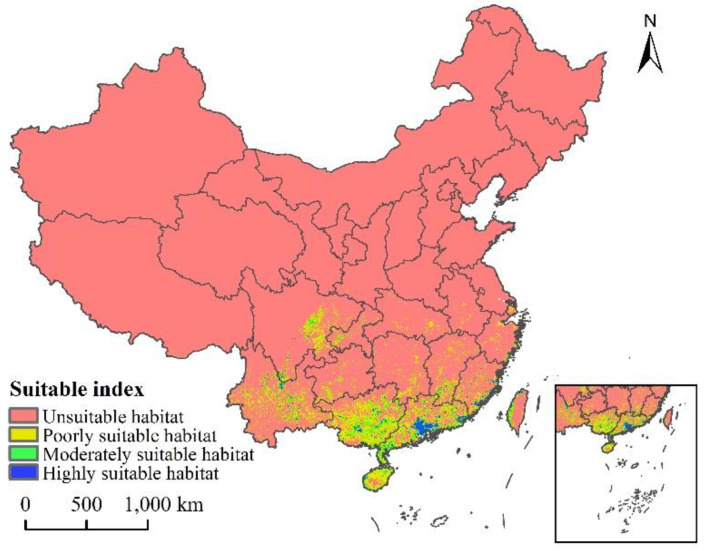
Potential current and suitable habitat for *L. invasa* in China.

**Figure 7 insects-12-00092-f007:**
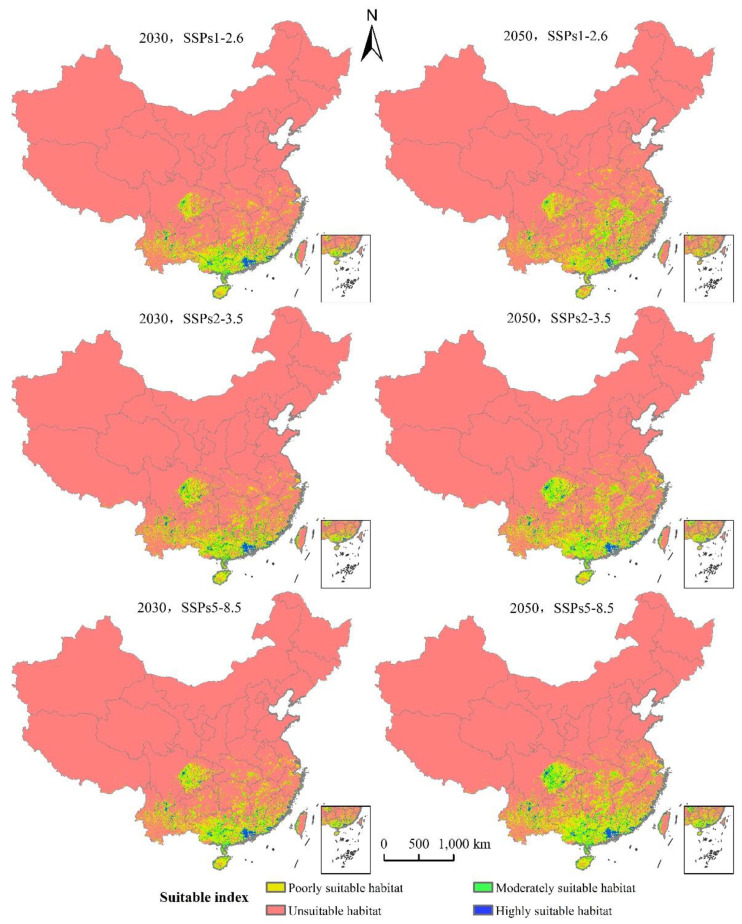
Potentially suitable climatic distribution of *L. invasa* under different climate change scenarios in China.

**Figure 8 insects-12-00092-f008:**
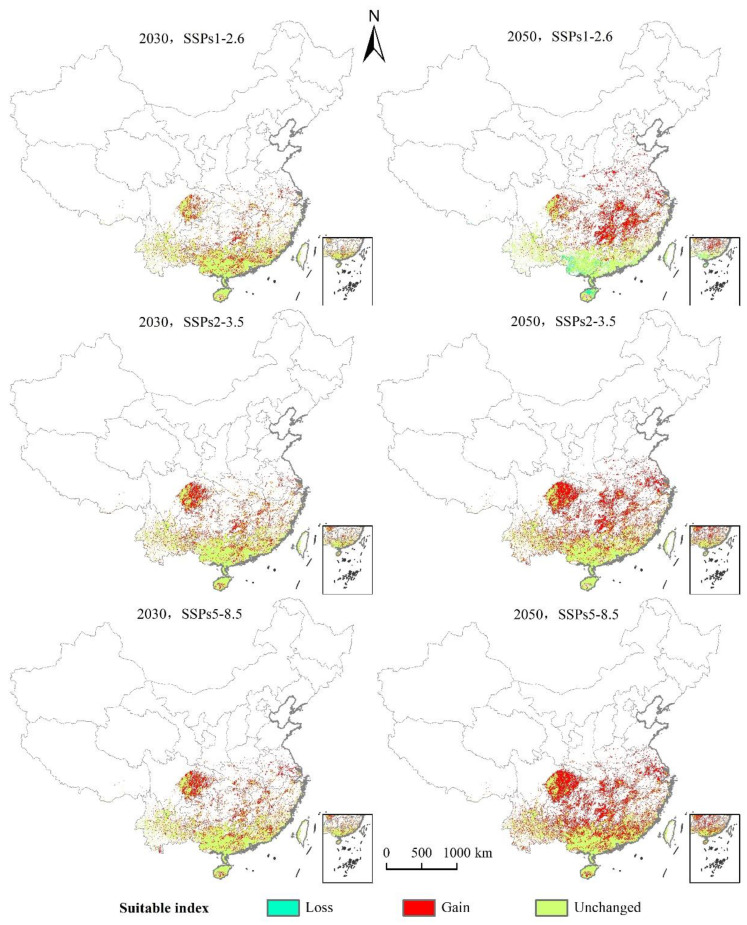
Changes in the potential geographical distribution of *L. invasa* under climate change scenarios.

**Figure 9 insects-12-00092-f009:**
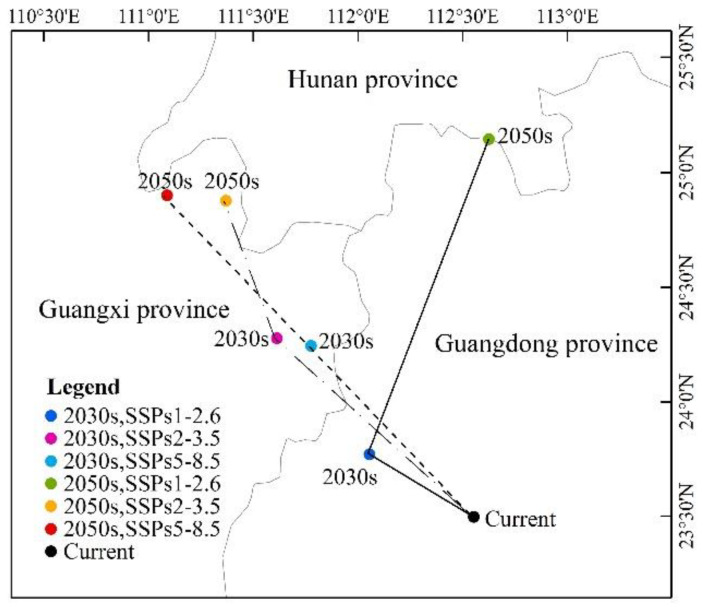
Highly suitable area centroid distributional shifts under climate change for *L. invasa*.

**Figure 10 insects-12-00092-f010:**
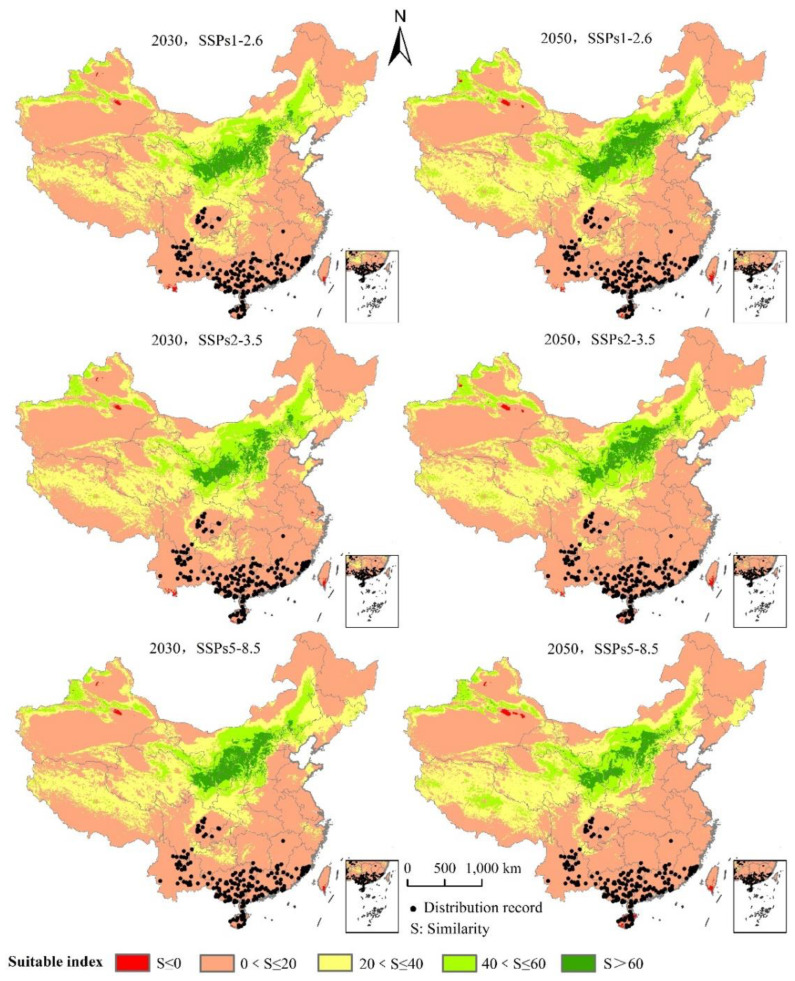
Analysis of the multivariate environmental similarity surface (MESS) of the potential area of distribution for *L. invasa* under future climate change.

**Table 1 insects-12-00092-t001:** Four emission scenarios.

Emission	Description
SSP1–2.6	SSP1 (Low forcing scenario) Upgrade to RCP2.6 scenario based on (Radiative forcing reaches 2.6 W/m^2^ in 2100)
SSP2–4.5	SSP2 (Medium forcing scenario) Upgrade to RCP4.5 scenario based on (Radiative forcing reaches 4.5 W/m^2^ in 2100)
SSP3–7.0	SSP3 (Medium forcing scenario) New RCP7.0 emission path based on (Radiative forcing will reach 7.0 W/m^2^ in 2100)
SSP5–8.5	SSP5 (High Forcing Scenario) Upgrade to RCP8.5 scenario based on (SSP5 is the only SSP scenario that can achieve radiative forcing to 8.5 W/m^2^ in 2100)

**Table 2 insects-12-00092-t002:** Environmental data used in the research.

Code	Description	Whether to Use for Modeling
Bio1	Annual Mean Temperature (℃)	Yes
Bio2	Mean Diurnal Range (Mean of monthly (max temp–min temp)) (℃)	No
Bio3	Isothermality (BIO2/BIO7) (×100)	Yes
Bio4	Temperature Seasonality (standard deviation × 100)	No
Bio5	Max Temperature of Warmest Month (℃)	No
Bio6	Min Temperature of Coldest Month (℃)	Yes
Bio7	Temperature Annual Range (BIO5-BIO6) (℃)	Yes
Bio8	Mean Temperature of Wettest Quarter (℃)	Yes
Bio9	Mean Temperature of Driest Quarter (℃)	Yes
Bio10	Mean Temperature of Warmest Quarter (℃)	No
Bio11	Mean Temperature of Coldest Quarter (℃)	Yes
Bio12	Annual precipitation (mm)	No
Bio13	Precipitation of Wettest Month (mm)	No
Bio14	Precipitation of Driest Month (mm)	No
Bio15	Precipitation Seasonality (Coefficient of Variation)	No
Bio16	Precipitation of Wettest Quarter (mm)	Yes
Bio17	Precipitation of Driest Quarter (mm)	No
Bio18	Precipitation of Warmest Quarter (mm)	Yes
Bio19	Precipitation of Coldest Quarter (mm)	NO
NDVI	Normalized Vegetation Index	Yes
Altitude	Altitude(m)	No

**Table 3 insects-12-00092-t003:** Suitable areas for *L. invasa* under different climate change scenarios (10^4^ km^2^).

Period	Highly Suitable	Moderately Suitable	Poorly Suitable	Total Suitable
Current	3.96	10.59	29.36	43.91
2030s, SSPs1–2.6	5.44	14.75	43.43	63.62
2030s, SSPs2–4.5	5.65	15.73	47.9	69.28
2030s, SSPs5–8.5	5.96	16.23	48.22	70.41
2050s, SSPs1–2.6	3.75	16.75	51.28	71.78
2050s, SSPs2–4.5	6.62	21.55	60.68	88.85
2050s, SSPs5–8.5	7.73	24.65	64.93	97.31

**Table 4 insects-12-00092-t004:** Future changes in suitable habitat area (10^4^ km^2^).

Period	Loss	Gain	Unchanged
2030s, SSPs1–2.6	0.01	17.68	40.74
2030s, SSPs2–3.5	0.002	22.80	40.73
2030s, SSPs5–8.5	0.002	24.08	40.73
2050s, SSPs1–2.6	2.15	28.58	38.60
2050s, SSPs2–3.5	0.05	40.95	40.70
2050s, SSPs5–8.5	0.002	48.07	40.73

## Data Availability

The data is included in the article. For the data provided in this study, see the titles “2.1. Distributional Data” and “2.2.1. Data Sources” in the text.

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
