# Peer review of "Predicting the Distribution of the Invasive Species *Leptocybe invasa*: Combining MaxEnt and Geodetector Models"

_insects, 2021, doi:10.3390/insects12020092_

Round 1

Reviewer 1 Report

I read with attention the manuscript entlited "Predicting the distribution of the invasive species Leptocybe invasa: combining  MaxEnt and Geodetector models". It is a very interesting paper about the prediction of the blue gum chalcid spreading in the light especially of climate changes.

I think the paper is well written, but it needs some little improvements and clarifications, following reported:

  • line 76: Did the authors refer to km or hm? I would suggest to use km
  • line 80: please add species information: author (Fisher & La Salle, 2004); order and family (Hymenoptera: Eulophidae).
  • line 90, 93 and whole manuscript: when already mentioned, please use L. invasa. This use is not allowed at the beginning of a Paragraph.
  • line 93: please move "The primary..." to a new line, it could stressed the paragraph of the "aims of the work".
  • line 119: what about the period 2001-2020?
  • 123-124: please change " the BCCCSM2-MR climate model" with "it"

Fig 4 legend: In my opinion there is a mistake, please rewrite.

Referring precisely to results reported in lines 251-258 ( figure 5): " the coldest monthly minimum temperature is - 33.78~17.50°C, the driest seasonal average temperature is -28.40 ~ 22.77°C", how did the author explain these results?  Taking into account that "the developmental zero temperature (DZT) for the complete life cycle (egg-adult) of L. invasa [...] was estimated to be about 19°C, while DZTs of egg, larval and pupal stages were 13, 19.7 and 17 °C, respectively" (please see: https://www.cabi.org/isc/datasheet/108923), could these temperature ranges be acceptable? Author should add some comments or opinion in discussion section.

Author Response

审稿人1

Reviewer 2 Report

The article is difficult to read because the text is poorly written (English language requires extensive revision)

The work is interesting, but the authors make a basic confusion between the species distribution models and the diffusive models. They are different models, with very different objectives and which require different data.
Furthermore, there is a methodological problem because the extension of China is such that the ranges of environmental variables in which the model is estimated are very different from the ranges where it is then applied.
With an extensive work of arranging the paper and perfecting the model it can be published (the topic is of interest)

Author Response

Reviewer 2

Reviewer 3 Report

The paper provides an interesting contribution to elucidate the importance of a model to predict a potential distribution of invasive Eucalyptus pest. Overall, the paper is well written. Methodology description would benefit by adding some more details, as suggested below. The overall length and data elaboration and literature choice are appropriate, tables are appropriate and self-explanatory.

The authors highlight how the use of models (MaxEnt and geographic detector) to predict the potential distribution of Leptocybe invasa. The investigated species is considered highly invasive and causes damage to Eucalyptus plants. Although the text is fluent and understandable, in this MS there is no information on the species examined.

  1. When did it arrive and where did it arrive in China?
  2. Has the spread of the species in the different regions been followed?
  3. The current distribution of Eucalyptus did not take into account in which Eucalyptus species is present and spread or if resistant eucalyptus species were already planted.
  4. Provide more information on the spreading capacity of the phytophagous species.

Furthermore, the authors do not believe that the recently introduced species is still in the process of spreading beyond global warming and temperature increases.

Graphs with distribution of Leptocybe invasa under different climate change scenarios are not clear to evidence changing. Could authors improve them? For example authors could highlight only the area where distribution will change.

Another MS has also been published on the same species (See doi/abs/10.1002/ps.5408) it would be appropriate for the authors to take this into account.

For a better understanding the authors are advised to write the abbreviations used in the abstract also in extended form

Line104-109: With what precision varied geographical position of the records obtained. This dataset represents the starting point for further analysis. 

Line 149-150The software used refers to this site Maxent (amnh.org)?

Line 255-258. The reported values are unclear. Thermal ranges are proposed or if we talk about average we propose a single value for example -28.40 ~ 22.77 °C

Line 383-385. The sentence is not clear the authors distinguish a stage of development or growth. They should refer to the life parameters of Leptocybe invasa. See: https://doi.org/10.1111/aen.12094

Author Response

审稿人3

Reviewer 4 Report

  1. The authors use the standard procedures of MAXENT for the modelling. However, there is no agreement in the literature on which set of parameter values to use in MaxEnt, and best practices suggest performing a preliminary sensitivity analysis on parameter performance for model selection (e.g., Hernandez et al 2006; Merow et al, 2013)"   

 Merow C, Smith MJ, Silander JA. A practical guide to MaxEnt for modeling species’ distributions: what it does, and why inputs and settings matter. Ecography. 2013; 36(10): 1058–1069

Hernandez PA, Graham CH, Master LL, Albert DL. The effect of sample size and species characteristics on performance of different species distribution modeling methods. Ecography. 2006; 29(5): 773– 785  

  1. The authors in explaining in point 2.2.3. the Data processing should consider to conduct first a correlation analyses between all the 21 environmental variables to exclude those that are autocorrelated.

  1. The authors use only AUC to validate the models. However, they should also use AIC ( see example at the following manuscript: https://journals.plos.org/plosone/article/comments?id=10.1371/journal.pone.0237216

  1. The authors should explain the ecological meaning of the selected variables for the target species.

Author Response

Reviewer 4

Round 2

Reviewer 2 Report

The authors were able to greatly improve the paper. From the authors' reply it does not seem that the revision suggestions have been well received, but the new version of the work nonetheless fills the weaknesses highlighted during the revision phase so the paper can be published

Author Response

Reviewer 2

Reviewer 3 Report

The authors answered the selected questions point by point. Small formatting aspects need to be considered.

I wish the best for their research

Author Response

Reviewer 3

Reviewer 4 Report

I am happy with the new version of the manuscript and for the explanations of the authors. Now the manuscript can be accepted.

Author Response

Reviewer 4
